# Test-Time Training with KV Binding Is Secretly Linear Attention

**Junchen Liu** [* 1 2 3]  **Sven Elflein** [1 2 3]  **Or Litany** [1 4]  **Zan Gojcic** [1]  **Ruilong Li** [* 1]

## Abstract

Test-time training (TTT) with KV binding as sequence modeling layer is commonly interpreted as a form of online meta-learning that memorizes a key–value mapping at test time. However, our analysis reveals multiple phenomena that contradict this memorization-based interpretation. Motivated by these findings, we revisit the formulation of TTT and show that a broad class of TTT architectures can be expressed as a form of learned linear attention operator. Beyond explaining previously puzzling model behaviors, this perspective yields multiple practical benefits: it enables principled architectural simplifications, admits fully parallel formulations that preserve performance while improving efficiency, and provides a systematic reduction of diverse TTT variants to a standard linear attention form. Overall, our results reframe TTT not as test-time memorization, but as learned linear attention with enhanced representational capacity. Project page: https://research.nvidia.com/labs/sil/projects/tttla/.

## 1. Introduction

Test-Time Training (TTT) has emerged as a powerful paradigm for dynamic model adaptation. Initially introduced to address distribution shift by updating model parameters on unlabeled test inputs, TTT has since evolved into a distinct architectural primitive (Sun et al., 2020). Recent work has increasingly framed TTT as an alternative to standard softmax attention in transformers, offering favorable properties like linear-time compute and constant memory usage during autoregressive inference (Zhang et al., 2025; Behrouz et al., 2026). Among the various TTT formulations,

this paper focuses on TTT with KV binding (Sun et al., 2025; Zhang et al., 2025; Han et al., 2025; Behrouz et al., 2026), which optimizes a self-supervised key-value association objective in the inner loop, as opposed to end-to-end methods that backpropagate from the final task loss (Tandon et al., 2025; Behrouz et al., 2025b). The prevailing interpretation of TTT casts it as a form of online meta-learning or memorization (Sun et al., 2025; Finn et al., 2017; Metz et al., 2018), where the "inner loop" dynamically constructs a temporary key-value (KV) map by optimizing a neural network (e.g., an MLP) on previously observed tokens. The subsequent inference step is viewed as querying this stored knowledge. This perspective has led to increased complexity in recent architectural designs, motivating the use of sophisticated optimizers, normalization schemes, and deep inner-loop networks (Zhang et al., 2025; Han et al., 2025; Behrouz et al., 2026; 2025a; Dalal et al., 2025), all explicitly designed to improve the fidelity of this "memorization".

**The Memorization Paradox.** Yet, this interpretation of TTT as "test-time memorization" can be directly contradicted by empirical evidence. If TTT would truly function by explicitly learning and retrieving key–value associations, its behavior would conform to basic principles of memory formation and optimization dynamics. Contrary to this expectation, we identify systematic anomalies that directly contradict this hypothesis:

- *Distributional Asymmetry.* Unlike standard attention, in which queries and keys share the same semantic space, converged TTT models exhibit a significant distributional mismatch between queries and keys.

- *Replacing queries with keys.* Replacing queries with keys for TTT models has negligible effect on the task performance, suggesting that queries do not play a functional retrieval role as in standard attention.

- *Optimization vs. Performance.* Counterintuitively, improvements in the inner loop, which can be interpreted as stronger "memorization", do not guarantee better downstream performance.

- *The Gradient Ascent Anomaly.* Most strikingly, we find that replacing inner-loop gradient descent with gradient ascent always preserves, and in some cases even improves, task performance.

These observations collectively challenge the prevailing

[*]Equal contribution [1]NVIDIA, Toronto, Ontario, Canada [2]University of Toronto, Toronto, Ontario, Canada [3]Vector Institute, Toronto, Ontario, Canada [4]Technion – Israel Institute of Technology, Haifa, Israel. Correspondence to: Ruilong Li <ruilongl@nvidia.com>.

*Proceedings of the 43rd International Conference on Machine Learning*, Seoul, South Korea. PMLR 306, 2026. Copyright 2026 by the author(s).

view of TTT as an online meta-learning or key-value memorization mechanism.

**TTT is Secretly Linear Attention.** Motivated by these observations, and by prior work showing that TTT reduces to linear attention in the restricted case of a single linear inner-loop layer with zero initialization (Sun et al., 2025), we revisit the mathematical formulation of TTT. We show analytically that even TTT variants with complex fast-weight parameterizations (including multi-layer MLPs and momentum) can be equivalently rewritten as a form of learned linear attention operator (Katharopoulos et al., 2020).

Under this unified view, the inner loop does not perform "meta learning" in the conventional sense. Instead, it induces a structured, history-dependent mixing of query, key, and value vectors. This perspective resolves the empirical paradoxes identified above: gradient ascent preserves performance because sign inversions are absorbed into the learned value projection, and distributional symmetry between queries and keys is unnecessary because the mechanism operates as a feature mixer rather than a similarity-based retrieval system.

**Practical Implications.** Unmasking TTT as Linear Attention is not just a theoretical exercise; it unlocks significant practical benefits. By adopting this perspective, we:

- *Simplify.* We show that many components introduced in prior TTT architectures, such as weight normalization and momentum, are often redundant.

- *Parallelize.* We derive a fully parallel form of TTT that achieves up to $4.0\times$ inference throughput on attention calculation while maintaining performance.

- *Generalize.* We provide a systematic reduction of diverse TTT variants to a common linear attention form. Empirically, the full TTT model exceeds this reduced linear-attention variant only marginally ($+0.87$ perplexity on LLM, $+0.24$ dB on NVS), suggesting that the additional inner-loop machinery contributes only modest gains beyond what standard linear attention already captures.

**Conflict of Interest Disclosure.** This work was conducted at NVIDIA. J.L., S.E., O.L., Z.G., and R.L. are employed by or affiliated with NVIDIA. The paper analyzes publicly described TTT architectures and evaluates open-source implementations rather than NVIDIA-developed products or systems. The authors do not report additional financial conflicts of interest.

## 2. Related Work

### 2.1. Linear Attention

Recurrent Neural Networks (RNNs) have recently gained renewed interest as efficient alternatives to standard Transformers (Vaswani et al., 2017). The introduction of linear attention (Katharopoulos et al., 2020) has accelerated advances in RNN-based architectures, leading to the incorporation of various mechanisms such as token-dependent decay factors (Gu et al., 2022; Smith et al., 2023; Orvieto et al., 2023; Peng et al., 2023; Sun et al., 2023) and, more recently, data-dependent decay (Qin et al., 2023; 2024; Peng et al., 2024; Gu & Dao, 2024; Dao & Gu, 2024; Zhang et al., 2024; Yang et al., 2024a). The data-dependent decay factor, termed the selective mechanism in Mamba (Gu & Dao, 2024; Dao & Gu, 2024), has been highlighted as crucial for strong in-context learning performance. DeltaNet (Schlag et al., 2021) conditions the update rule on both the current token and state for improved retrieval, and chunk-parallelization (Yang et al., 2024b) has enabled its efficient deployment in many recent architectures (Yang et al., 2024a; Peng et al., 2025a; Behrouz et al., 2026; Zhong et al., 2025; Team et al., 2025; Peng et al., 2025b; Lei et al., 2025; Liu et al., 2025). Notably, DeltaNet and its variants are equivalent to TTT with a single linear layer and MSE loss (Yang et al., 2024a).

### 2.2. Test-Time Training

Test-time training (TTT) broadly refers to methods that continue to update model parameters during inference. This concept was first introduced to address train-test distribution shift (Sun et al., 2020; Gandelsman et al., 2022), where models adapt to test data by optimizing a self-supervised objective at inference time. Subsequently, TTT has been explored for improving inference-time performance in specific applications such as 3D reconstruction (Chen et al., 2024; 2025; Yuan et al., 2025). More recently, TTT has been developed as a sequence modeling architecture with linear complexity, serving as an alternative to softmax attention in transformers (Sun et al., 2025; Wang et al., 2025; Zhang et al., 2025; Han et al., 2025; Dalal et al., 2025; Behrouz et al., 2026).

When used as a sequence modeling layer, TTT has two main variants: (1) methods that use a key-value binding loss (e.g., dot-product or MSE loss) as the inner-loop objective (Sun et al., 2025; Zhang et al., 2025; Han et al., 2025; Behrouz et al., 2026), referred to as TTT-KVB in prior work (Tandon et al., 2025), and (2) methods that perform end-to-end back-propagation through the inner loop from the final task loss (e.g., cross-entropy in language modeling), referred to as TTT-E2E (Tandon et al., 2025; Behrouz et al., 2025b). This paper focuses on the first variant.

TTT has demonstrated effectiveness across diverse tasks, including language modeling (Sun et al., 2025; Wang et al., 2025; Zhang et al., 2025), video generation (Dalal et al., 2025; Zhang et al., 2025), novel view synthesis (Zhang et al., 2025), and image classification (Han et al., 2025). In this paradigm, part of the model parameters (known as

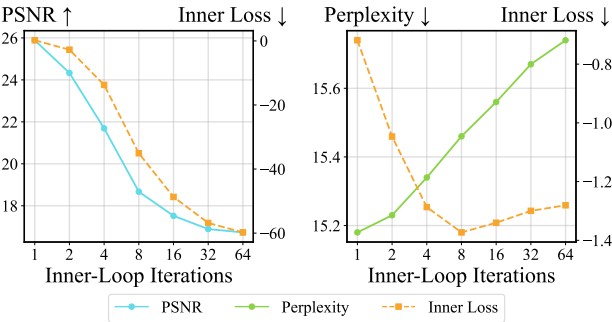

*Figure 1.* **Inner-Loop Optimization vs. Performance**. Increasing inner-loop iterations improves inner-loop loss but degrades task performance, contradicting the memorization-based interpretation of TTT. Experiments are based on LaCT (Zhang et al., 2025).

fast weights (Hinton & Plaut, 1987)) are updated online at test time using a self-supervised key-value association loss, with the goal of memorizing history associations. Since this nested optimization is performed in-context, TTT is also referred to as in-context meta-learning (Finn et al., 2017; Metz et al., 2018) and is tightly related to the concept of fast weight programming (Schlag et al., 2021) and test-time scaling (Muennighoff et al., 2025; Snell et al., 2024).

The design space of TTT is rich and has been actively explored. LaCT (Zhang et al., 2025) improves hardware utilization through large chunk sizes. Based on the key-value memorization perspective, other works have explored advanced test-time optimizers (Behrouz et al., 2026; Zhang et al., 2025; Karami et al., 2025) and alternative regression targets (Han et al., 2025; Behrouz et al., 2025c). Notably, prior work has shown that when the inner loop consists of a single linear layer, TTT can be reinterpreted as a linear attention operator (Sun et al., 2025). In this work, we demonstrate that this interpretation extends to general TTT architectures with complex multi-layer MLPs as inner loops, and show that this perspective enables practical benefits including simplified formulations and efficient parallel implementations.

## 3. Preliminary

We provide a brief overview of the TTT mechanism that this paper focuses on. In TTT, each sequence modeling layer maintains a set of fast weights $f_\theta$ (typically a lightweight MLP) that is updated *during both training and inference*. Given an input sequence, tokens are first projected into keys $K$, values $V$, and queries $Q$ (similar to standard attention). The core idea is to perform *online gradient descent* on $f_\theta$ using a self-supervised key-value binding objective: for each token, the key $k$ serves as input and the value $v$ serves as the regression target, i.e., $\mathcal{L} = \|f_\theta(k) - v\|^2$ (or a dot-product loss variant). After updating $\theta$ with this objective, the query $q$ is passed through the updated function $f_\theta$ to produce the output.

This mechanism is commonly interpreted as a *storage-and-retrieval* system (Sun et al., 2025; Finn et al., 2017): the inner-loop optimization "memorizes" key-value associations into $f_\theta$, which are later "retrieved" by querying the learned function. Under this view, architectural capacity, optimizer selection, and the number of inner-loop steps are all motivated by achieving more faithful memorization of key–value associations.

This variant of test-time training, which optimizes a key-value binding objective in the inner loop, is referred to as TTT-KVB in prior work (Tandon et al., 2025). Among the various TTT formulations (Sun et al., 2025; Gandelsman et al., 2022; Sun et al., 2020; Tandon et al., 2025; Behrouz et al., 2025b), this paper exclusively focuses on this variant.

## 4. Empirical Contradictions to Memorization

As we show in this section, the empirical behavior of TTT models consistently violates the properties implied by the storage-and-retrieval interpretation.

### 4.1. Better Inner Loss Leads to Worse Performance

Under a memorization-based interpretation, inner-loop loss serves as a natural proxy for memorization quality: lower loss indicates more accurate encoding of key–value mappings and should therefore improve task performance.

To test this property, we vary the number of inner-loop gradient steps at inference time, which usually yields lower inner-loop loss in pretrained TTT models (Zhang et al., 2025; Han et al., 2025), without changing the architecture, training data, or learned parameters.

Figure 1 shows that despite improved inner-loop fitting, downstream performance degrades consistently as the number of inner-loop steps increases. This inverse relationship holds across both LLMs and novel view synthesis (NVS) tasks. Such behavior directly contradicts the memorization-based interpretation of TTT, under which more accurate key–value fitting should be beneficial, or at least not harmful for the downstream performance. Instead, these results indicate that the inner loop affects model computation in a fundamentally different manner, inconsistent with conventional notions of test-time memory.

### 4.2. TTT with Gradient Ascent

The degradation in downstream performance with improved inner-loop fitting raises a more fundamental question: *is gradient-descent-based memorization in the inner loop necessary at all?*

To test this, we replace gradient descent in the inner loop with *gradient ascent*, effectively flipping the sign of all fast-weight gradients. Importantly, this is not a pure inference-

*Table 1.* **Observations Contradicting the Storage-and-Retrieval Interpretation of TTT.** Replacing gradient descent with ascent breaks the storage interpretation, and replacing queries with keys breaks the retrieval interpretation, yet task performance remains mostly unchanged. Experiments are base on LaCT (Zhang et al., 2025) and ViTTT (Han et al., 2025).

| Model | Perplexity ↓ (LaCT-LLM) | PSNR ↑ (LaCT-NVS) | Top-1 Acc↑ (ViTTT) |
|---|---|---|---|
| Baseline | 16.43 | 25.94 | 79.34 |
| Gradient Ascent | 16.19 | 25.85 | 79.61 |
| Replace $Q$ with $K$ | 16.18 | 25.95 | 79.18 |

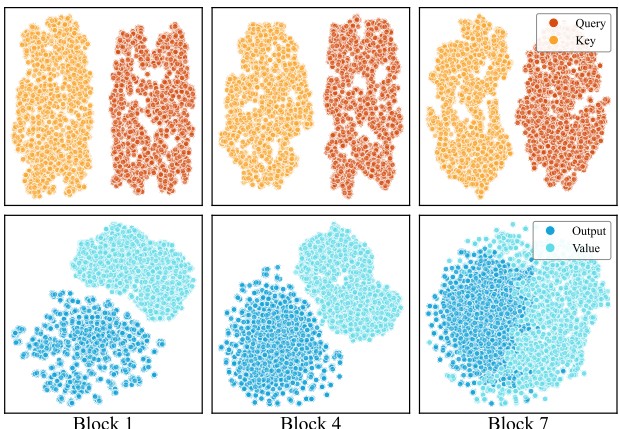

Block 1  Block 4  Block 7

*Figure 2.* **Distributional Asymmetry Between $Q$ and $K$.** t-SNE visualizations of $(Q, K)$ and $(V, O)$ features in a pretrained LaCT (Zhang et al., 2025) model on the NVS task, showing that the TTT inner loop is evaluated out of distribution and thus does not perform reliable retrieval.

time intervention on a pretrained checkpoint: each gradient-ascent model is retrained from scratch with the sign-flipped inner-loop update applied throughout both training and inference, so the surrounding parameters have a chance to adapt to the inverted objective. Under a memorization-based interpretation, this change should still have a strong negative effect, as gradient ascent explicitly worsens the fit to the key–value regression objective.

Contrary to this expectation, across all evaluated models and tasks, TTT with gradient ascent performs comparably to, and in some cases even slightly better than, standard gradient descent (Table 1). This holds despite the fact that gradient ascent consistently increases the inner-loop loss.

Notably, in methods such as LaCT (Zhang et al., 2025), where the inner-loop loss is a Frobenius inner product, flipping the gradient is equivalent to negating the loss itself. The fact that TTT remains effective even under such an inverted objective provides additional evidence that TTT does not rely on memorizing a key–value mapping.

### 4.3. Distributional Asymmetry Between $Q$ and $K$

As we mentioned above, TTT is commonly interpreted as a *storage-and-retrieval* mechanism. In the previous subsections, we showed that the *storage* aspect of this interpretation is empirically inconsistent with model behavior. We now examine whether the *retrieval* aspect provides a plausible explanation.

During the TTT inner-loop update, the parametric function (e.g., MLP) is optimized using keys $K$ as inputs and values $V$ as regression targets. After this update, queries $Q$ are feed into the same function to produce outputs $O$. For such a retrieval process to be effective, the distribution of $Q$ must be close to that of the $K$ used during optimization. Otherwise, the parametric function is evaluated out-of-distribution, and its outputs are not expected to reliably lie on the manifold of $V$. In other words, meaningful retrieval requires substantial distributional overlap between $Q$ and $K$.

To test this assumption, we analyze the distributions of $Q$ and $K$ in a pretrained LaCT (Zhang et al., 2025) model on the NVS task. For each layer, we collect $Q$ and $K$ vectors across all tokens and visualize their distributions using t-SNE (Maaten & Hinton, 2008). As shown in Figure 2, we observe a pronounced and consistent mismatch between $Q/K$ and $V/O$ across layers, indicating that the parametric function optimized on keys, is systematically evaluated on out-of-distribution inputs when applied to queries, at both training and inference time. Under such conditions, the resulting outputs cannot be interpreted as reliable retrieval of stored key–value information, and this mismatch persists regardless of how accurately the inner loop fits the key–value regression objective.

### 4.4. Replacing $Q$ with $K$

The distributional asymmetry between queries and keys again raises a more direct question: *is the query representation $Q$ necessary at all for TTT to function?*

To test this, we perform a simple experiment in which we replace the query $Q$ with the key $K$ when computing the TTT output. In standard attention mechanisms, such a substitution would be highly disruptive. Because attention weights depend on query–key similarity, replacing $Q$ with $K$ typically leads to degenerate behavior dominated by self-similarity and a substantial drop in performance.

Contrary to the expectation, we observe little to no degradation in TTT's performance. For both LaCT and ViTTT formulations performance remains comparable (Table 1). This insensitivity to the query representation contradicts a retrieval-based interpretation of TTT. If the inner loop relied on query-based access to stored information, replacing $Q$ with $K$ should significantly alter model behavior. Instead, the observed invariance indicates that $Q$ does not function

as a meaningful query into a memorized key-value map.

**Summary.** Taken together, these observations challenge the interpretation of test-time training as a storage-and-retrieval mechanism. Downstream behavior is largely insensitive to both the quality and direction of inner-loop optimization, and to the presence of a meaningful query signal. These failures suggest that in practice TTT does not operate as test-time memorization.

## 5. TTT Is Secretly Linear Attention

If TTT is not functioning as a memorization mechanism, how should its behavior be understood? An important clue comes from prior work, which shows that in the restricted setting of a single linear inner-loop layer with zero initialization, TTT is exactly equivalent to linear attention (Sun et al., 2025). More generally, despite their apparent complexity, existing TTT variants share the defining computational properties of linear attention: linear-time computation and a constant-size state with respect to sequence length.

Motivated by this connection, we re-examine TTT by explicitly unrolling the inner-loop updates. We show analytically that TTT induces a linear attention–like operator in a general form, even when the inner loop consists of multi-layer non-linear mappings (Section 5.1). This perspective naturally explains our empirical observations that contradict the storage-and-retrieval interpretation (Section 5.2). Finally, we rewrite two representative variants of TTT, LaCT (Zhang et al., 2025) and ViTTT (Han et al., 2025), in their linear attention form (Sections 5.3 and 5.4). Detailed derivations are deferred to the appendix.

### 5.1. General Form

**Theorem 5.1** (Linearization of Inner-Loop Updates). *Consider a TTT model whose inner-loop function has a linear, bias-free final layer,*

$$f(x) = \phi(x; \Theta)\, W,$$

*where $\phi(x; \Theta) \in \mathbb{R}^{D_\mathrm{h}}$ denotes the hidden representation of the inner-loop function with parameters $\Theta$, and $W \in \mathbb{R}^{D_\mathrm{h} \times D_\mathrm{out}}$ is the weight matrix of the final layer. Suppose that at step $t$, the inner loop performs one step of gradient descent on an objective $\mathcal{L}$ with learning rate $\eta$, using key input $k$, updating all trainable parameters,*

$$(W_{t+1}, \Theta_{t+1}) = (W_t, \Theta_t) - \eta \nabla_{(W_t, \Theta_t)} \mathcal{L}(f_t(k)),$$

*where $\phi_t(\cdot) \triangleq \phi(\cdot; \Theta_t)$ and $f_t(\cdot) \triangleq \phi_t(\cdot) W_t$. Then, for any query $q$, the output after the update can be written as*

$$o = \phi_{t+1}(q) \left( W_t + \phi_t(k)^\top g_t(k) \right), \quad g_t(k) \triangleq -\eta \frac{\partial \mathcal{L}}{\partial f_t(k)}.$$

*This expression is a linear attention operator of the form*

$$o = \hat{q} \left( S_0 + \hat{k}^\top \hat{v} \right),$$

*where*

$$\hat{q} = \phi_{t+1}(q), \quad \hat{k} = \phi_t(k), \quad \hat{v} = g_t(k), \quad S_0 = W_t.$$

The complete proof is provided in Appendix B.

**Theorem 5.2** (Unrolling Inner-Loop Updates). *Given a sequence of query–key pairs $\{(q_0, k_0), (q_1, k_1), \ldots, (q_t, k_t)\}$, suppose the TTT model performs one gradient descent step per input in sequence. By repeated application of Theorem 5.1, the parameters after processing token $t$ are*

$$(W_{t+1}, \Theta_{t+1}) = (W_0, \Theta_0) - \eta \sum_{i=0}^{t} \nabla_{(W_i, \Theta_i)} \mathcal{L}(f_i(k_i)).$$

*Evaluating the TTT model on query $q_t$ yields*

$$o_t = \phi_{t+1}(q_t)\, W_{t+1}$$
$$= \phi_{t+1}(q_t) \left( W_0 + \sum_{i=0}^{t} \phi_i(k_i)^\top g_i(k_i) \right),$$

*where $g_i(k_i)$ is defined as in Theorem 5.1. This corresponds to the extended linear attention form on sequential inputs.*

$$o_t = \hat{q}_t \left( S_0 + \sum_{i=0}^{t} \hat{k}_i^\top \hat{v}_i \right).$$

The complete proof is provided in Appendix C.

Next, we extend Theorem 5.2 to the case where the inner-loop employs gradient descent with momentum.

**Theorem 5.3** (Gradient Descent with Momentum). *Given the momentum-augmented gradient accumulator defined as*

$$(\Delta W_t, \Delta \Theta_t) = \nabla_{(W, \Theta)} \mathcal{L}(f_t(k_t)) + \alpha_t (\Delta W_{t-1}, \Delta \Theta_{t-1}),$$

*where $\alpha_t$ denotes the (possibly token-dependent) momentum factor at step $t$. The model parameters are then updated according to*

$$(W_{t+1}, \Theta_{t+1}) = (W_t, \Theta_t) - \eta (\Delta W_t, \Delta \Theta_t).$$

*Define the cumulative momentum coefficient as*

$$\beta_i^j \triangleq \begin{cases} \prod_{s=i+1}^{j} \alpha_s & \text{if } i < j, \\ 1 & \text{if } i = j. \end{cases}$$

*Unrolling this recurrence and evaluating the TTT model on query $q_t$ yields*

$$o_t = \phi_{t+1}(q_t)\, W_{t+1}$$
$$= \phi_{t+1}(q_t) \left( W_0 + \sum_{i=0}^{t} \phi_i(k_i)^\top m_i(k_i) \right),$$

*which induces a linear-attention–like form identical to that of Theorem 5.2, with the effective value vector being a momentum-weighted sum:*

$$\hat{v}_i = m_i(k_i) \triangleq g_i(k_i) \cdot \sum_{j=i}^{t} \beta_i^j.$$

The complete proof is provided in Appendix D.

### 5.2. Explanation of TTT Empirical Behaviors

Having shown that TTT can be rewritten as a linear attention operator (Theorem 5.1 - 5.3), we can now revisit the empirical behaviors that appeared contradictory under the memorization-based interpretation. The linear-attention perspective provides a unified and mechanistic explanation.

**More Inner-Loop Steps.** According to Theorem 5.1, the inner loop does not perform storage of key–value information, but instead defines a learnable mapping that transforms the original inputs into the effective query, key, and value representations. This mapping depends on inner-loop hyperparameters, including the number of optimization steps. Increasing the number of inner-loop iterations at inference time therefore induces an attention operator different to the one used during training, naturally leading to degraded performance due to train–test mismatch rather than improved memorization.

**Gradient Ascent in the Inner Loop.** Under the same formulation, replacing gradient descent with gradient ascent simply flips the sign of the effective value vector $g_t$. Since this sign is absorbed into the learned attention operator and the mapping itself is optimized under the downstream objective, the model adapts to this change. This explains why TTT remains effective under gradient ascent, despite the absence of a meaningful memorization objective.

**Distributional Asymmetry Between $Q$ and $K$.** The linear-attention view clarifies why similarity between $q$ and $k$ is not required. In TTT, $q$ and $k$ influence different components of the attention operator: $q$ determines the effective query via $\phi_{t+1}(q)$, while $k$ determines the effective key and value via $\phi_t(k)$ and $g_t(k)$, respectively. They are therefore intermediate features rather than symmetric query–key representations, making distributional mismatch expected rather than pathological.

**Replacing $Q$ with $K$.** Finally, replacing $q$ with $k$ does not collapse the attention mechanism because the effective query and key remain distinct: $\phi_{t+1}(k)$ versus $\phi_t(k)$. Since $\phi$ is learnable and evaluated at different parameter states, the model can map the same input to different representations, preserving attention functionality.

**Summary.** Viewed through the lens of linear attention, the inner loop of TTT parameterizes a structured linear-form attention operator rather than performing test-time storage and retrieval. Under this perspective, the observed empirical behaviors follow naturally from representation learning and train–test consistency considerations.

### 5.3. Example: LaCT as Linear Attention

We now show how a representative instantiation of TTT formula, LaCT (Zhang et al., 2025), can be rewritten in the form of linear attention.

LaCT adopts a bias-free SwiGLU MLP (Shazeer, 2020) as its inner-loop mapping, parameterized by three learnable weight matrices $W_0, W_2 \in \mathbb{R}^{D_h \times D_k}$ and $W_1 \in \mathbb{R}^{D_h \times D_v}$:

$$f(x) = \big(\mathrm{silu}(xW_0) \odot (xW_2)\big)W_1.$$

The inner-loop objective is defined via the Frobenius inner product

$$\mathcal{L}(f(k), v) = -\langle f(k), v \rangle.$$

At each token, LaCT performs gradient descent with per-token learning rate $\eta_t$, momentum $\alpha_t$, and gradient orthogonalization $\mathcal{M}(\cdot)$ inspired by Muon (Jordan et al., 2024):

$$W_{i,t+1} = W_{i,t} - \eta_t \, \mathcal{M}(\Delta W_{i,t}),$$

$$\Delta W_{i,t} = \nabla_{W_{i,t}} \mathcal{L}(f_t(k_t), v_t),$$

for $i \in \{0, 1, 2\}$.

Following Theorem 5.3, the inner-loop model at step $t$ can be written as

$$f_t(x) = \phi_t(x)\, W_{1,t}, \quad \phi_t(x) = \mathrm{silu}(xW_{0,t}) \odot (xW_{2,t}).$$

Evaluating the updated model on query $q_t$ yields

$$
\begin{aligned}
o_t &= f_{t+1}(q_t) \\
&= \phi_{t+1}(q_t)\, W_{1,t+1} \\
&= \phi_{t+1}(q_t) \left( W_{1,0} + \sum_{i=0}^{t} \mathcal{M}\big(\phi_i(k_i)^\top m_i\big) \right), \quad (1)
\end{aligned}
$$

where

$$m_i(k_i) \triangleq v_i \cdot \sum_{j=i}^{t} \eta_j \, \beta_i^j.$$

This expression reveals that the inner loop of LaCT is effectively a linear attention–like operator, with $\phi_i(k_i)$ and $m_i(k_i)$ playing the roles of keys and values, and $\phi_{t+1}(q_t)$ acting as the query vector. Note that LaCT also applies weight normalization after each update, which we omit here for simplicity. See Appendix E for more details.

## 5.4. Example: ViTTT as Linear Attention

We next show that another instantiation of test-time training, ViTTT (Han et al., 2025), can likewise be rewritten in the form of linear attention.

ViTTT employs fast weights consisting of two independent components: (i) a simplified gated linear unit (GLU), and (ii) a depthwise convolution layer. These components are updated independently in the inner loop. We show that each admits a linear-attention interpretation, implying that ViTTT as a whole falls within the same framework.

**GLU component.** The GLU is defined as

$$f(x) = \text{silu}(xW_0) \odot (xW_1),$$

where $W_0$ and $W_1$ are fast weights updated via gradient descent. As in LaCT, the inner-loop loss is defined using a Frobenius inner product. Following a derivation analogous to previous sections (see Appendix F), evaluating the updated GLU on a query $q_t$ yields

$$\phi(x) = \text{silu}(xW_0),$$

$$o_t = \phi_{t+1}(q_t) \odot \Big( q_t \big( W_1 + k_t^\top (v_t \odot \phi_t(k_t)) \big) \Big).$$

This expression takes the form of linear attention, with $\phi_t(k_t)$ acting as a multiplicative gate on values and $\phi_{t+1}(q_t)$ gating the final output.

**Depthwise convolution component.** ViTTT additionally includes a $3 \times 3$ depthwise convolution layer with fast weights, updated in the inner loop. As convolution is effectively sliding window linear layer, this TTT component is equivalent to a sliding window linear attention. A formal derivation is provided in Appendix G.

**Discussion.** Since both fast-weight components in ViTTT admit linear-attention formulations, their combination also induces a linear-attention–like operator. This establishes ViTTT as another concrete instance of test-time training whose behavior is more naturally understood through the lens of linear attention.

# 6. Practical Implications

Interpreting TTT through the lens of linear attention is not merely a theoretical exercise but yields concrete practical benefits. This perspective reveals a systematic trajectory for reducing complex TTT formulations into linear attention, along which we observe that several commonly adopted design choices, such as per-token learnable learning rates, weight normalization, are not essential to final performance. By clarifying which components are truly valuable and which are unnecessary, this view enables substantial simplification of TTT formulations. Moreover, recognizing

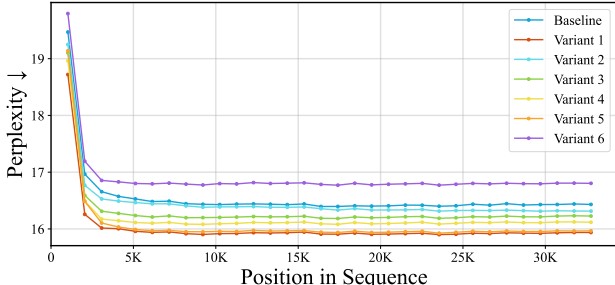

*Figure 3.* **Perplexity Metric for Ablation on LaCT-LLM.** Evaluated on 2.5B tokens from the Book-3 dataset.

TTT as linear attention makes explicit that the seemingly recurrent inner-loop updates admit a parallel formulation, leading to significant efficiency gains in both training and inference. All experiments in this section use the official implementations of LaCT (Zhang et al., 2025) for the LLM and NVS tasks, and ViTTT (Han et al., 2025) for the image recognition task. Detailed experimental settings are provided in Appendix A.

## 6.1. Reduce TTT to Linear Attention

Viewing TTT through the lens of linear attention reveals that several design choices that were justified under a storage-and-retrieval interpretation are in fact redundant or optional. This perspective enables a principled ablation path that progressively simplifies complex TTT variants such as LaCT and ViTTT into standard linear attention. Below, we outline this reduction trajectory and analyze the role of each component. The ablation steps are applied *sequentially*: Variant 1 is the Baseline with Step 1 applied, Variant 2 adds Step 2 on top of Variant 1, and so on, so that Variant 6 corresponds to the full reduction from the original TTT formulation to standard linear attention.

**Step 1. Update only the last-layer parameters.** As shown in Theorem 5.1, the effective query and key vectors are $\phi_{t+1}(q)$ and $\phi_t(k)$, where $\phi_t(\cdot) \triangleq \phi(\cdot; \Theta_t)$. Updating $\Theta_t$ within the inner loop makes $\phi_t$ a dynamic kernel that is difficult to unroll analytically. If the inner loop instead updates only the final-layer parameter, $\Theta$ remains fixed during TTT, and $\phi(\cdot) \triangleq \phi(\cdot; \Theta)$ becomes a static function with learnable parameters. From the linear-attention perspective, $\phi(\cdot)$ then acts as a learnable kernel function with effective queries and keys given by $\phi(q)$ and $\phi(k)$.

**Step 2. Remove weight normalization.** LaCT applies weight normalization to all learnable parameters $\Theta_t, W_t$ after each inner-loop update. After Step 1, normalization on $\Theta_t$ is a no-op since $\Theta_t$ is fixed. Normalizing the final-layer parameter $W_t$, which corresponds to the state in the linear attention view, is therefore equivalent to normalizing the state $S_t$ (see Appendix E.4). As such normalization is uncommon in linear attention literature, we remove it

*Table 2.* **Ablation Trajectory Reducing TTT to Standard Linear Attention.** By progressively simplifying complex TTT formulations into standard linear attention (Variant 6), we quantify the contribution of each design component in TTT. Variant 1 achieves the best performance across all three tasks. Variants 2–6 admit parallel implementations. We additionally report the inference throughput of each variant's TTT layer on the LLM task. † denotes ablations that do not apply, in which case performance matches the preceding variant. * indicates that ViTTT does not use gradient orthogonalization, so we ablate gradient normalization instead.

| Alias | Description of TTT Inner-loop Designs | Perplexity ↓ (LaCT-LLM) | PSNR ↑ (LaCT-NVS) | Top-1 Acc↑ (ViTTT) | Tokens Per Sec↑ (Recurrent Impl.) | Tokens Per Sec↑ (Parallel Impl.) |
|---|---|---|---|---|---|---|
| Baseline | LaCT (Zhang et al., 2025) / ViTTT (Han et al., 2025) | 16.43 | 25.94 | 79.34% | 4.30M | N/A |
| Variant 1 | Baseline w/ only update the last layer parameters | **15.93** | **25.97** | **79.63%** | 10.60M | N/A |
| Variant 2 | Variant 1 w/ remove weight normalization | 16.31 | 25.93 | 79.63%† | 11.02M | 30.18M |
| Variant 3 | Variant 2 w/ multi-layer MLP → singel linear layer | 16.23 | 25.71 | 79.39% | 12.95M | 49.69M |
| Variant 4 | Variant 3 w/ remove per-token learnable lr | 16.12 | 25.70 | 79.39%† | 13.31M | 53.99M |
| Variant 5 | Variant 4 w/ remove momentum in SGD | 15.97 | 25.70† | 79.39%† | 14.40M | 57.28M |
| Variant 6 | Variant 5 w/ remove gradient orthogonalization | 16.80 | 25.73 | 79.54%* | **89.67M** | **124.6M** |

for ablation. Notably, after this step the TTT formulation becomes fully parallelizable, as discussed in Section 6.2.

**Step 3. Multi-layer MLP → single linear layer.** Several TTT variants (Han et al., 2025; Behrouz et al., 2026) employ deeper inner-loop MLPs, but report inconsistent empirical gains. From a linear attention perspective, increasing MLP depth simply induces a more complex kernel function $\phi(\cdot)$ over queries and keys. When $q$ and $k$ already have sufficient representational capacity, this added complexity is unlikely to help. We therefore reduce the multi-layer MLP to a single linear layer, effectively removing $\phi(\cdot)$ altogether. This exposes the true queries and keys as $\hat{q} = q$ and $\hat{k} = k$, bringing the formulation closer to basic linear attention.

**Step 4. Remove per-token learning rates.** Many TTT methods (Behrouz et al., 2026; Zhang et al., 2025) introduce a per-token learnable learning rate $\eta_t$. As shown in Section 5.3, with Frobenius dot product as inner loss this can be absorbed into the learnable $v_t$, indicating it's functionally redundant. Consistent with our finding, ViTTT empirically finds that a constant learning rate of 1.0 suffices.

**Step 5. Remove momentum in SGD.** As shown in Theorem 5.3, adding momentum to the inner-loop SGD update, as done in LaCT and related methods (Behrouz et al., 2026), only alters the effective value $\hat{v}$, from the instantaneous gradient $g_t(k)$ to a momentum-weighted sum of past gradients. From the linear attention perspective, this corresponds to remixing historical key–value contributions into a single value vector. Since both keys and values are already learnable, this additional mixing is unlikely to provide meaningful benefit, thus we remove momentum for ablation. Notably, for both LaCT and ViTTT, $g_t(k) = -v$ and removing momentum recovers $\hat{v} = v$, exposing the true value vector.

**Step 6. Remove gradient orthogonalization.** As discussed in Section 5.3, LaCT optionally applies gradient orthogonalization $\mathcal{M}(\Delta W)$. Under the linear attention reformulation, this corresponds to applying an operator to the state update $\mathcal{M}(\hat{k}^\top v)$. We remove this operation for ablation. After this final step, Both LaCT and ViTTT reduce

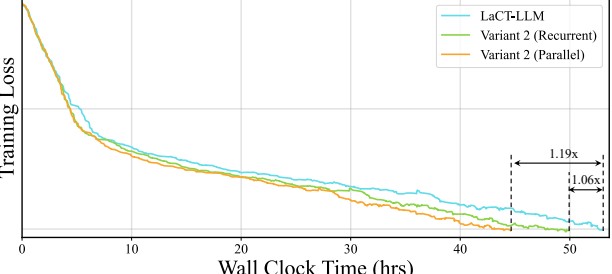

*Figure 4.* **Training loss vs. wall-clock time on LaCT-LLM.** We compare the original LaCT-TTT with both parallel and recurrent form of Variant 2. The parallel form achieves a $1.19\times$ end-to-end speedup while maintaining comparable convergence.

exactly to standard linear attention: $o = q\left(W + \sum_i k_i^\top v_i\right)$.

**Results.** We progressively apply the above ablations to LaCT on the LLM and NVS tasks, and to ViTTT on the image recognition task, with results summarized in Table 2. Surprisingly, restricting the inner loop to update only the final MLP layer consistently yields the best overall performance across tasks, suggesting that many of the more complex design choices are unnecessary, or even detrimental. Most other components contribute only marginally to performance, with two notable exceptions: deeper MLPs are beneficial for the NVS task, while gradient orthogonalization improves performance on the LLM task. Overall, reducing the full TTT formulation to a basic linear attention operator (Variant 6) results in only minor performance degradation ($+0.4$ perplexity on LLM and $-0.2$ dB on NVS). For the LLM task, Table 2 reports perplexity at 32k sequence length, with results across other lengths shown in Figure 3.

**6.2. Parallel Form of TTT**

Existing TTT variants are typically implemented in a recurrent manner, reflecting their original storage-and-retrieval interpretation. However, as we have shown that TTT can alternatively be viewed as a form of linear attention, a natural question arises: can TTT admit a parallel formulation that enables more efficient implementation? We show that under certain conditions, corresponding to Variants 2–6 in our ablation, such a parallel form indeed exists.

The key insight is that when weight normalization is removed and only the final-layer parameters are updated, the state update becomes associative. In this setting, the kernel function $\phi_t(\cdot) \triangleq \phi(\cdot; \Theta_t)$ is static and independent of sequence history, allowing the recurrence in Theorem 5.3 to be computed via a parallel prefix scan rather than sequential token-by-token updates.

We implement this parallel formulation for LaCT on the LLM task. As shown in Table 2, switching from the recurrent to the parallel implementation improves the inference throughput of the TTT layer by up to $4.0\times$ (measured in tokens per second, single batch). Combined with the simplifications from Step 1 and Step 2, this yields a $1.19\times$ end-to-end training speedup without degrading model quality, as shown in Figure 4.

We provide the full parallel formulation and a proof of equivalence to the sequential recurrence in Appendix H. In Appendix I, we further show that introducing weight normalization or dynamic kernel functions breaks associativity, thereby preventing parallelization.

## 7. Conclusion

In this work, we challenge the prevailing view of TTT as a mechanism for test-time memorization of key–value mappings. Through systematic empirical analysis, we identify several anomalies—including gradient ascent behavior, distributional asymmetry between queries and keys, and the lack of correlation between inner-loop convergence and downstream performance—that are fundamentally incompatible with a memorization-based interpretation.

We provide an alternative explanation by showing that TTT, even with complex inner loops involving multi-layer MLPs and momentum-based optimizers, can be analytically rewritten as a form of linear attention operator. Under this view, the inner loop does not perform meta learning in the conventional sense, but instead parameterizes a structured mixing of queries, keys, and values. Viewing TTT through the lens of linear attention not only resolves the observed paradoxes, but also enables principled simplifications, parallel implementations with improved efficiency, and a unified framework for understanding TTT variants.

Our analysis applies broadly to TTT formulations based on key–value binding, but is limited to settings where the inner-loop final layer is linear and bias-free. Empirically, we focus on LaCT (Zhang et al., 2025) and ViTTT (Han et al., 2025) as case studies because they are representative methods with open-source implementations. Other key–value-binding variants such as Titans (Behrouz et al., 2026) and Atlas (Behrouz et al., 2025a) also satisfy the assumptions of our theorems, and we expect our results to extend to them; empirical validation on these models is left to future

work as their implementations become available. Extending these insights to nonlinear final layers, and exploring deeper connections between TTT and modern linear attention mechanisms in both directions, remain additional avenues for future work.

## Acknowledgements

We gratefully acknowledge the Vector Institute for providing compute resources for part of this work. We thank Tianyuan Zhang for sharing experimental details and insightful discussions, and Yu Sun for reviewing an early version of this paper.

## Impact Statement

Our work simplifies and parallelizes existing TTT architectures, potentially reducing the computational cost and energy consumption of training and deploying such models. Because the contributions are primarily analytical and efficiency-oriented, we do not foresee additional societal risks beyond those already associated with the underlying sequence modeling architectures and their downstream applications.

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

# A. Experiment Setup

Here we outline the three tasks used to evaluate the TTT framework, along with the experimental setup.

**Language Modeling**  We use the 760M parameter LaCT-LLM (Zhang et al., 2025) as our baseline model. All models are trained on 100B tokens sampled from the FineWeb-Edu dataset (Penedo et al., 2024) with a batch size of 4 per GPU across 8 NVIDIA A100 GPUs for 20K iterations, which takes approximately 56 hours to complete. All other hyperparameters follow the original (Zhang et al., 2025) configuration. For evaluation, we report perplexity on 2.5B tokens from the Book-3 dataset (Gao et al., 2020). All implementations are based on the Flame (Zhang & Yang, 2025) codebase.

**Novel View Synthesis**  We use 12-layer 768 hidden dimension LaCT-NVS (Zhang et al., 2025) model as baseline, totaling 114M parameters. We train our model on the RealEstate10K (Zhou et al., 2018) dataset with a batch size of 128 per GPU across 4 NVIDIA A100 GPUs for 20K iterations, which takes approximately 38 hours to complete. We use 2 input views and 6 target views for training, and 2 input views and 3 target views for evaluation. All images are resized to $128 \times 128$ resolution. We train using only the MSE loss and evaluate using the Peak Signal-to-Noise Ratio (PSNR) metric. All other hyperparameters follow the original LaCT-NVS (Zhang et al., 2025) configuration.

**Image Classification**  We use ViTTT-B (Han et al., 2025) as our baseline model, totaling 90M parameters. Follow (Han et al., 2025), we train our model on the ImageNet-1K (Deng et al., 2009) dataset with a batch size of 256 per GPU across 2 NVIDIA H100 GPUs for 60 epochs, which takes approximately 16 hours to complete. We evaluate on the ImageNet-1K (Deng et al., 2009) validation set using the top-1 accuracy. All other hyperparameters follow the original (Han et al., 2025) configuration.

# B. Proof of Theorem 5.1

*Proof.* We explicitly unroll the single-step inner-loop update. Using the shorthand notation $\phi_t(\cdot) \triangleq \phi(\cdot; \Theta_t)$ and $f_t(\cdot) \triangleq \phi_t(\cdot) W_t$, the model output for the key input $k$ at step $t$ is

$$f_t(k) = \phi_t(k)\, W_t.$$

By the chain rule, the gradient of the last layer with respect to the loss $\mathcal{L}(f_t(k))$ is

$$\nabla_{W_t}\mathcal{L} = \phi_t(k)^\top \frac{\partial \mathcal{L}}{\partial f_t(k)}.$$

Applying one step of gradient descent with learning rate $\eta$ yields

$$W_{t+1} = W_t - \eta \phi_t(k)^\top \frac{\partial \mathcal{L}}{\partial f_t(k)}$$
$$= W_t + \phi_t(k)^\top g_t(k)$$

where we define

$$g_t(k) \triangleq -\eta \frac{\partial \mathcal{L}}{\partial f_t(k)} \in \mathbb{R}^{D_{\text{out}}}.$$

The kernel function parameters $\Theta$ are also updated with gradient descent, yielding the updated function $\phi_{t+1}(\cdot) = \phi(\cdot; \Theta_{t+1})$.

Evaluating the updated model on a query $q$ gives

$$o = \phi_{t+1}(q)\, W_{t+1}$$
$$= \phi_{t+1}(q) \left( W_t + \phi_t(k)^\top g_t(k) \right)$$

This expression is a linear attention operator of the form

$$o = \hat{q} \left( S_0 + \hat{k}^\top \hat{v} \right),$$

where

$$\hat{q} = \phi_{t+1}(q), \quad \hat{k} = \phi_t(k), \quad \hat{v} = g_t(k), \quad S_0 = W_t.$$

$\square$

## C. Proof of Theorem 5.2

*Proof.* We prove by induction on the number of tokens processed.

**Base case.** Starting from initial parameters $(W_0, \Theta_0)$, processing token 0 with key $k_0$ applies one gradient descent step. By Theorem 5.1:

$$W_1 = W_0 + \phi_0(k_0)^\top g_0(k_0),$$

where $g_0(k_0) = -\eta \frac{\partial \mathcal{L}}{\partial f_0(k_0)}$, and $\Theta_1$ is updated accordingly. This matches the claimed form with $t = 0$.

**Inductive step.** Assume that after processing tokens $0, \ldots, t-1$, we have

$$W_t = W_0 + \sum_{i=0}^{t-1} \phi_i(k_i)^\top g_i(k_i).$$

Processing token $t$ with key $k_t$ applies another gradient descent step. By Theorem 5.1:

$$W_{t+1} = W_t + \phi_t(k_t)^\top g_t(k_t).$$

Substituting the inductive hypothesis:

$$W_{t+1} = W_0 + \sum_{i=0}^{t-1} \phi_i(k_i)^\top g_i(k_i) + \phi_t(k_t)^\top g_t(k_t) = W_0 + \sum_{i=0}^{t} \phi_i(k_i)^\top g_i(k_i).$$

**Output computation.** Evaluating the model on query $q_t$ with the updated parameters gives:

$$o_t = \phi_{t+1}(q_t) W_{t+1}$$

$$= \phi_{t+1}(q_t) \left( W_0 + \sum_{i=0}^{t} \phi_i(k_i)^\top g_i(k_i) \right)$$

This is the extended linear attention form:

$$o_t = \hat{q}_t \left( S_0 + \sum_{i=0}^{t} \hat{k}_i^\top \hat{v}_i \right),$$

where $\hat{q}_t = \phi_{t+1}(q_t)$, $\hat{k}_i = \phi_i(k_i)$, $\hat{v}_i = g_i(k_i)$, and $S_0 = W_0$. □

## D. Proof of Theorem 5.3

*Proof.* We prove by induction, extending the approach of Theorem 5.2 to momentum updates.

**Base case.** At $t = 0$, the momentum accumulator is initialized as $\Delta W_{-1} = 0$, so

$$\Delta W_0 = \nabla_W \mathcal{L}(f_0(k_0)) = \phi_0(k_0)^\top \frac{\partial \mathcal{L}}{\partial f_0(k_0)}.$$

Applying the update rule $W_1 = W_0 - \eta \Delta W_0$ gives

$$W_1 = W_0 + \phi_0(k_0)^\top g_0(k_0),$$

where $g_0(k_0) = -\eta \frac{\partial \mathcal{L}}{\partial f_0(k_0)}$. This matches the claimed form with $m_0(k_0) = g_0(k_0)$.

**Inductive step.** Define the cumulative momentum coefficient from step $i$ to step $j$ as:

$$\beta_i^j \triangleq \begin{cases} \prod_{s=i+1}^{j} \alpha_s & \text{if } i < j, \\ 1 & \text{if } i = j. \end{cases}$$

Assume that after processing tokens $0, \ldots, t-1$, the momentum accumulator satisfies

$$\Delta W_{t-1} = \sum_{i=0}^{t-1} \beta_i^{t-1} \, \phi_i(k_i)^\top \frac{\partial \mathcal{L}}{\partial f_i(k_i)}.$$

At step $t$, the momentum update gives

$$\Delta W_t = \nabla_W \mathcal{L}(f_t(k_t)) + \alpha_t \Delta W_{t-1}$$

$$= \phi_t(k_t)^\top \frac{\partial \mathcal{L}}{\partial f_t(k_t)} + \alpha_t \sum_{i=0}^{t-1} \beta_i^{t-1} \, \phi_i(k_i)^\top \frac{\partial \mathcal{L}}{\partial f_i(k_i)}.$$

For terms $i \in [0, t-1]$, we have $\alpha_t \cdot \beta_i^{t-1} = \alpha_t \prod_{s=i+1}^{t-1} \alpha_s = \prod_{s=i+1}^{t} \alpha_s = \beta_i^t$. For the $i = t$ term, the coefficient is $\beta_t^t = 1$. Thus:

$$\Delta W_t = \sum_{i=0}^{t} \beta_i^t \, \phi_i(k_i)^\top \frac{\partial \mathcal{L}}{\partial f_i(k_i)}.$$

**Weight accumulation.** From $W_{j+1} = W_j - \eta \Delta W_j$, we have $W_{t+1} = W_0 - \eta \sum_{j=0}^{t} \Delta W_j$. Substituting the closed form for $\Delta W_j$ and exchanging the order of summation:

$$W_{t+1} = W_0 - \eta \sum_{j=0}^{t} \sum_{i=0}^{j} \beta_i^j \, \phi_i(k_i)^\top \frac{\partial \mathcal{L}}{\partial f_i(k_i)}$$

$$= W_0 + \sum_{i=0}^{t} \phi_i(k_i)^\top \underbrace{\left( -\eta \frac{\partial \mathcal{L}}{\partial f_i(k_i)} \sum_{j=i}^{t} \beta_i^j \right)}_{m_i(k_i)}.$$

The momentum-weighted effective value is thus

$$m_i(k_i) = g_i(k_i) \cdot \sum_{j=i}^{t} \beta_i^j.$$

**Output computation.** Evaluating the model on query $q_t$ with the updated parameters:

$$o_t = \phi_{t+1}(q_t) \, W_{t+1}$$

$$= \phi_{t+1}(q_t) \left( W_0 + \sum_{i=0}^{t} \phi_i(k_i)^\top m_i(k_i) \right).$$

This is the linear attention form with $\hat{q}_t = \phi_{t+1}(q_t)$, $\hat{k}_i = \phi_i(k_i)$, $\hat{v}_i = m_i(k_i)$, and $S_0 = W_0$. $\qquad \square$

# E. Derivation: LaCT as Linear Attention

We provide the full derivation showing how LaCT (Zhang et al., 2025) can be written in the form of linear attention, following the framework of Theorem 5.3.

### E.1. LaCT Architecture and Update Rule

LaCT adopts a bias-free SwiGLU MLP as its inner-loop mapping:

$$f(x) = \big(\mathrm{silu}(xW_0) \odot (xW_2)\big)W_1,$$

where $W_0, W_2 \in \mathbb{R}^{D_h \times D_k}$ and $W_1 \in \mathbb{R}^{D_h \times D_v}$. The inner-loop objective uses the Frobenius inner product:

$$\mathcal{L}(f(k), v) = -\langle f(k), v \rangle.$$

At each token $t$, LaCT performs gradient descent with per-token learning rate $\eta_t$, momentum $\alpha_t$, and Muon-style (Jordan et al., 2024) gradient orthogonalization $\mathcal{M}(\cdot)$:

$$\Delta W_{i,t} = \nabla_{W_{i,t}}\mathcal{L}(f_t(k_t), v_t) + \alpha_t \Delta W_{i,t-1}, \quad W_{i,t+1} = W_{i,t} - \eta_t\, \mathcal{M}(\Delta W_{i,t}),$$

for $i \in \{0, 1, 2\}$.

### E.2. Gradient Computation

The upstream gradient with respect to the output is $\frac{\partial \mathcal{L}}{\partial f_t(k_t)} = -v_t$. Using the chain rule, the gradient for the final layer $W_1$ is:

$$\nabla_{W_1}\mathcal{L} = \phi_t(k_t)^\top \frac{\partial \mathcal{L}}{\partial f_t(k_t)} = -\phi_t(k_t)^\top v_t,$$

where $\phi_t(x) = \mathrm{silu}(xW_{0,t}) \odot (xW_{2,t})$ is the kernel function.

### E.3. Linear Attention Form with Muon

Following the same induction as Theorem 5.3, but with the Muon orthogonalization $\mathcal{M}(\cdot)$ applied after each gradient accumulation, the weight update becomes:

$$W_{1,t+1} = W_{1,0} - \sum_{j=0}^{t} \eta_j\, \mathcal{M}(\Delta W_{1,j}).$$

Using the cumulative momentum coefficient $\beta_i^t$ defined in Theorem 5.3, the momentum accumulator for $W_1$ is:

$$\Delta W_{1,t} = -\phi_t(k_t)^\top v_t + \alpha_t \Delta W_{1,t-1} = -\sum_{i=0}^{t} \beta_i^t\, \phi_i(k_i)^\top v_i.$$

Substituting into the weight update and exchanging the order of summation (as in the proof of Theorem 5.3):

$$W_{1,t+1} = W_{1,0} + \sum_{i=0}^{t} \mathcal{M}\big(\phi_i(k_i)^\top m_i\big),$$

where the momentum-weighted effective value is:

$$m_i \triangleq v_i \cdot \sum_{j=i}^{t} \eta_j\, \beta_i^j.$$

Evaluating on query $q_t$:

$$o_t = \phi_{t+1}(q_t)\, W_{1,t+1}$$
$$= \phi_{t+1}(q_t)\left(W_{1,0} + \sum_{i=0}^{t} \mathcal{M}\big(\phi_i(k_i)^\top m_i\big)\right).$$

This is a linear attention–like form where $\phi_i(k_i)$ and $m_i$ play the roles of keys and values, and $\phi_{t+1}(q_t)$ acts as the query. The Muon orthogonalization $\mathcal{M}(\cdot)$ is applied element-wise to each key-value outer product before accumulation.

### E.4. Effect of Weight Normalization

LaCT additionally applies weight normalization to $W_1$ after each update:

$$W_{1,t+1} = \text{Norm}(W_{1,t} - \eta_t \, \mathcal{M}(\Delta W_{1,t})),$$

where $\text{Norm}(\cdot)$ applies channel-wise $\ell_2$ normalization.

Weight normalization does *not* break the linear attention perspective. Recall that in the linear attention formulation, the state matrix $S_t = W_{1,t}$ accumulates key-value outer products. With weight normalization, we simply normalize this state after each update:

$$S_{t+1} = \text{Norm}\big(S_t + \phi_t(k_t)^\top m_t\big).$$

The output remains a linear function of the (normalized) state:

$$o_t = \phi_{t+1}(q_t) \, S_{t+1}.$$

Thus, LaCT with weight normalization is still a linear attention mechanism—the query linearly reads from a state that accumulates key-value information.

However, the normalization prevents expressing the state as a *simple sum* over history. Unlike the unnormalized case where $S_{t+1} = S_0 + \sum_{i=0}^{t} \phi_i(k_i)^\top m_i$, the nested normalization creates a sequential dependency:

$$S_{t+1} = \text{Norm}\big(\text{Norm}\big(S_{t-1} + \phi_{t-1}(k_{t-1})^\top m_{t-1}\big) + \phi_t(k_t)^\top m_t\big).$$

This has implications for parallelization, which we discuss in Appendix I.

## F. Derivation: ViTTT GLU as Linear Attention

We show that the simplified GLU used in ViTTT (Han et al., 2025) can be written into a linear attention form with element-wise multiplication.

### F.1. Architecture and Loss

In ViTTT, the GLU is defined as:

$$f_t(x) = \text{silu}(xW_{0,t}) \odot (xW_{1,t}),$$

where $W_{0,t}, W_{1,t} \in \mathbb{R}^{D_h \times D_h}$ are both square matrices. We define the kernel function as

$$\phi_t(x) \triangleq \text{silu}(xW_{0,t}).$$

Unlike the general form where the kernel function output is followed by matrix multiplication with a final layer, here the GLU uses element-wise multiplication between the nonlinear gating $\phi_t(x)$ and the linear projection $xW_{1,t}$.

The inner-loop loss uses the Frobenius inner product:

$$\mathcal{L}(f_t(k_t), v_t) = -\langle f_t(k_t), v_t \rangle.$$

### F.2. Gradient Computation

The upstream gradient with respect to the GLU output is $\frac{\partial \mathcal{L}}{\partial f_t(k_t)} = -v_t$.

Applying the chain rule, the gradient of $W_{1,t}$ is:

$$\nabla_{W_{1,t}} \mathcal{L} = k_t^\top \left( \frac{\partial \mathcal{L}}{\partial f_t(k_t)} \odot \phi_t(k_t) \right) = -k_t^\top (v_t \odot \phi_t(k_t)).$$

### F.3. Linear Attention Form

After one step of gradient descent with learning rate $\eta$:

$$W_{1,t+1} = W_{1,t} - \eta \nabla_{W_{1,t}} \mathcal{L} = W_{1,t} + \eta \, k_t^\top (v_t \odot \phi_t(k_t)).$$

Similarly, $W_{0,t}$ is updated to $W_{0,t+1}$, yielding the updated kernel function $\phi_{t+1}(\cdot)$.

Evaluating the model on query $q_t$ gives:

$$
\begin{aligned}
o_t = f_{t+1}(q_t) &= \phi_{t+1}(q_t) \odot (q_t W_{1,t+1}) \\
&= \phi_{t+1}(q_t) \odot \left( q_t W_{1,t} + \eta \, (q_t k_t^\top)(v_t \odot \phi_t(k_t)) \right) \\
&= \phi_{t+1}(q_t) \odot \left( q_t \big(W_{1,t} + \eta \, k_t^\top (v_t \odot \phi_t(k_t))\big) \right).
\end{aligned}
$$

This is a linear attention form where:

- The second term computes a scalar attention weight $\langle q_t, k_t \rangle$ that modulates the value vector $v_t$

- $\phi_t(k_t)$ acts as a multiplicative gate on the values

- $\phi_{t+1}(q_t)$ gates the final output

The state matrix $S_t = W_{1,t}$ accumulates outer products of keys and gated values, consistent with the general linear attention framework.

## G. Derivation: ViTTT Depthwise Convolution as Linear Attention

We show that the $3 \times 3$ depthwise convolution layer in ViTTT (Han et al., 2025) can be formulated as a form of spatially-local linear attention.

### G.1. Architecture and Loss

Let $K, V \in \mathbb{R}^{C \times H \times W}$ be the spatial key and value tensors, and $W_t \in \mathbb{R}^{C \times 1 \times 3 \times 3}$ be the depthwise convolution weights at step $t$. The forward pass computes:

$$f_t(K) = \mathrm{Conv}_{3 \times 3}(K; W_t),$$

where $\mathrm{Conv}_{3 \times 3}$ denotes depthwise convolution with $3 \times 3$ kernel.

The inner-loop loss uses the Frobenius inner product:

$$\mathcal{L}(f_t(K), V) = -\langle f_t(K), V \rangle = -\sum_{c,i,j} [f_t(K)]_{c,i,j} \cdot V_{c,i,j}.$$

### G.2. Gradient Computation

The upstream gradient is $\frac{\partial \mathcal{L}}{\partial f_t(K)} = -V$.

For depthwise convolution, the gradient with respect to the weight $W_t$ can be written as:

$$\nabla_{W_t} \mathcal{L} = K \star \frac{\partial \mathcal{L}}{\partial f_t(K)} = -K \star V,$$

where $\star$ denotes cross-correlation. Specifically, for each channel $c$ and offset $(\delta_y, \delta_x) \in \{-1, 0, 1\}^2$:

$$[\nabla_{W_t} \mathcal{L}]_{c, \delta_y, \delta_x} = -\sum_{i,j} K_{c, i+\delta_y, j+\delta_x} \cdot V_{c,i,j}.$$

### G.3. Linear Attention Form

After one step of gradient descent with learning rate $\eta$:

$$[W_{t+1}]_{c,\delta_y,\delta_x} = [W_t]_{c,\delta_y,\delta_x} + \eta \sum_{i,j} K_{c,i+\delta_y,j+\delta_x} \cdot V_{c,i,j}.$$

When evaluating on query $Q \in \mathbb{R}^{C \times H \times W}$, the output at position $(i,j)$ is:

$$
\begin{aligned}
O_{c,i,j} &= \sum_{\delta_y,\delta_x} [W_{t+1}]_{c,\delta_y,\delta_x} \cdot Q_{c,i+\delta_y,j+\delta_x} \\
&= \underbrace{\sum_{\delta_y,\delta_x} [W_t]_{c,\delta_y,\delta_x} \cdot Q_{c,i+\delta_y,j+\delta_x}}_{[\mathrm{Conv}(Q;W_t)]_{c,i,j}} \\
&\quad + \eta \sum_{i',j'} \underbrace{\Big( \sum_{\delta_y,\delta_x} Q_{c,i+\delta_y,j+\delta_x} \cdot K_{c,i'+\delta_y,j'+\delta_x} \Big)}_{\text{spatial attention weight}} \cdot V_{c,i',j'}.
\end{aligned}
$$

This is a linear attention form where:

- The first term corresponds to the initial state $S_0 = W_t$

- The attention weight between query position $(i,j)$ and key position $(i',j')$ is the sum of element-wise products over the $3 \times 3$ neighborhood offsets

- This spatially-local attention allows each output position to attend to all key-value positions, weighted by the overlap of their local $3 \times 3$ neighborhoods

Conceptually, since convolution is effectively a sliding-window linear layer, this TTT component is equivalent to a sliding-window linear attention mechanism.

## H. Parallel Form of TTT

We present the parallel formulation for TTT when only $W_1$ is dynamic (while $W_0$ and $W_2$ are static) and weight normalization is omitted. Under these conditions, the kernel function becomes static:

$$\phi(x) = \mathrm{silu}(xW_0) \odot (xW_2),$$

and the state update is associative, enabling parallel computation.

### H.1. Parallel Formulation

The parallel formulation operates on a sequence of $N$ chunks, each of size $L$. Let $\mathbb{Q}, \mathbb{K} \in \mathbb{R}^{(NL) \times D_k}$ and $\mathbb{V} \in \mathbb{R}^{(NL) \times D_v}$ denote the concatenated query, key, and value inputs across all chunks. Define the batched kernel function:

$$\Phi(\mathbf{X}) = \mathrm{silu}(\mathbf{X}W_0) \odot (\mathbf{X}W_2) \in \mathbb{R}^{(NL) \times D_h}.$$

To express chunk-wise operations, we introduce:

- Block-diagonal matrix $\mathbf{B} \in \mathbb{R}^{(NL) \times (NL)}$: $\mathbf{B} = \mathrm{diag}(I_L, \ldots, I_L)$ with $N$ identity blocks

- Per-chunk learning rate $\boldsymbol{\eta} \in \mathbb{R}^N$ and per-chunk momentum coefficient $\boldsymbol{\alpha} \in \mathbb{R}^N$

- Effective accumulation mask $\mathcal{C} \in \mathbb{R}^{N \times N}$ that folds the per-chunk learning rates into the cumulative momentum decay:

$$\mathcal{C}_{ti} = \sum_{j=i}^{t} \eta_j \prod_{s=i+1}^{j} \alpha_s \quad \text{for } t \geq i, \quad 0 \text{ otherwise.}$$

The output can be computed in parallel as:

$$\mathbb{O} = \Phi(\mathbb{Q}) W_{1,0} + \left( (\Phi(\mathbb{Q}) \Phi(\mathbb{K})^\top) \odot \mathcal{C}^{\uparrow L} \right) \mathbb{V},$$

where $(\cdot)^{\uparrow L}$ denotes the Kronecker product with $\mathbf{1}_{L \times L}$ to expand the $N \times N$ mask to $(NL) \times (NL)$.

### H.2. Proof of Equivalence

We prove that this parallel formulation is equivalent to the sequential recurrence.

*Proof.* Consider the sequential formulation where at each chunk $t$, the weight update is:

$$\Delta W_t = \nabla_{W_1} \mathcal{L}(f_t(K_t)) + \alpha_t \Delta W_{t-1}, \quad W_{1,t+1} = W_{1,t} - \eta_t \Delta W_t.$$

Since $\nabla_{W_1} \mathcal{L} = \Phi(K_t)^\top \frac{\partial \mathcal{L}}{\partial f_t(K_t)}$ and using the Frobenius inner product loss, we have:

$$\Delta W_t = -\Phi(K_t)^\top V_t + \alpha_t \Delta W_{t-1}.$$

**Step 1: Unroll the momentum recurrence.** Expanding $\Delta W_t$:

$$\Delta W_t = -\sum_{i=0}^{t} \left( \prod_{s=i+1}^{t} \alpha_s \right) \Phi(K_i)^\top V_i.$$

**Step 2: Unroll the weight recurrence.** From $W_{1,t+1} = W_{1,0} - \sum_{j=0}^{t} \eta_j \Delta W_j$, substituting and rearranging:

$$W_{1,t+1} = W_{1,0} + \sum_{i=0}^{t} \Phi(K_i)^\top V_i \cdot \underbrace{\sum_{j=i}^{t} \eta_j \prod_{s=i+1}^{j} \alpha_s}_{c_i^t}.$$

**Step 3: Compute the sequential output.** The output at chunk $t$ is:

$$O_t = \Phi(Q_t) W_{1,t+1} = \Phi(Q_t) W_{1,0} + \sum_{i=0}^{t} \Phi(Q_t) \Phi(K_i)^\top V_i \cdot c_i^t.$$

**Step 4: Match with parallel formulation.** By construction, the mask $\mathcal{C}^{\uparrow L}$ at block $(t, i)$ has value $\mathcal{C}_{ti} = \sum_{j=i}^{t} \eta_j \prod_{s=i+1}^{j} \alpha_s = c_i^t$ for $i \leq t$. For any token position $p$ in chunk $t$, the parallel form therefore computes:

$$[\mathbb{O}]_p = \Phi(Q_t)_p W_{1,0} + \sum_{i=0}^{t} \Phi(Q_t)_p \Phi(K_i)^\top V_i \cdot c_i^t,$$

which exactly matches the sequential output. $\square$

## I. Non-Reducible Case Analysis

In Section 6.2, we showed that TTT becomes reducible (and thus parallelizable) when only $W_1$ is dynamic while $W_0$ and $W_2$ are static. Here, we analyze two cases that break reducibility: (1) updating the kernel function parameters $\Theta = \{W_0, W_2\}$, and (2) applying weight normalization.

### I.1. Case 1: Dynamic Kernel Function (Updating $W_0$ and $W_2$)

When $W_0$ and $W_2$ are also updated, the kernel function $\phi_t(x) = \mathrm{silu}(xW_{0,t}) \odot (xW_{2,t})$ becomes history-dependent. Consider the gradient update for $W_0$ (the analysis for $W_2$ is analogous):

$$\nabla_{W_{0,t}}\mathcal{L} = K_t^\top \Big( \frac{\partial \mathcal{L}}{\partial f_t(K_t)} W_{1,t}^\top \odot (K_t W_{2,t}) \odot \mathrm{silu}'(K_t W_{0,t}) \Big).$$

The weight update becomes:

$$W_{0,t+1} = W_{0,t} - \eta_t \nabla_{W_{0,t}}\mathcal{L}.$$

Expanding recursively, the kernel function at step $t$ depends on $W_{0,t}$:

$$\phi_t(K_t) = \mathrm{silu}(K_t W_{0,t}) \odot (K_t W_{2,t}).$$

Substituting the recursive expression for $W_{0,t}$, the kernel function involves nested nonlinearities:

$$\phi_t(K_t) = \mathrm{silu}\Big( K_t \big(W_{0,t-1} - \eta_{t-1} \nabla_{W_{0,t-1}}\mathcal{L}\big)\Big) \odot (\cdots),$$

where $\nabla_{W_{0,t-1}}\mathcal{L}$ itself contains $\mathrm{silu}'(K_{t-1}W_{0,t-1})$.

The nested $\mathrm{silu}$ and $\mathrm{silu}'$ functions create a non-linear dependency chain: computing $\phi_t$ requires $W_{0,t}$, which depends on $\mathrm{silu}'(K_{t-1}W_{0,t-1})$, which in turn depends on $W_{0,t-1}$, and so on. This nested structure prevents expressing the output as a simple weighted sum over history, breaking the associativity required for parallel prefix scan.

### I.2. Case 2: Weight Normalization

Even when only $W_1$ is dynamic, applying weight normalization (as in LaCT, see Appendix E) introduces non-reducibility for the parallel formulation.

As discussed in Appendix E, weight normalization does not break the linear attention *interpretation*—it simply normalizes the state $S_t$ after each token. However, it does prevent the *parallel computation* enabled by the sum form.

The key issue is that normalization is not associative:

$$\mathrm{Norm}(A + B) \neq \mathrm{Norm}(A) + \mathrm{Norm}(B). \tag{2}$$

The parallel formulation in Section 6.2 relies on expressing the state as $S_{t+1} = S_0 + \sum_{i=0}^{t} \phi(K_i)^\top m_i$, which can be computed via associative parallel prefix scan. With weight normalization, the state becomes:

$$S_{t+1} = \mathrm{Norm}\Big( \mathrm{Norm}(S_{t-1} + \phi(K_{t-1})^\top m_{t-1}) + \phi(K_t)^\top m_t \Big).$$

This nested structure creates a strict sequential dependency: computing $S_{t+1}$ requires the fully normalized $S_t$, which requires $S_{t-1}$, and so on. The associativity required for parallel prefix scan is broken, forcing token-by-token sequential computation even though the underlying mechanism remains linear attention.

