# OpenReview forum: "Test-Time Training with KV Binding Is Secretly Linear Attention"
_ICML.cc/2026/Conference — ICML 2026 regular_

### Official Review · Reviewer_k31F · 2026-03-07

**Soundness:** 3
**Presentation:** 2
**Significance:** 3
**Originality:** 2
**Overall Recommendation:** 4
**Confidence:** 3

**Summary:**

The paper revisits the interpretation of Test-Time Training (TTT) architectures and argues that many TTT formulations can be rewritten as linear attention operators. Empirical observations—such as robustness to gradient ascent in the inner loop and insensitivity to replacing queries with keys—challenge the common interpretation of TTT as a test-time memorization mechanism. The authors derive a theoretical formulation showing that TTT architectures with fast-weight updates can be expressed as linear attention, and demonstrate that this view enables architectural simplifications and parallel implementations.

**Compliance With Llm Reviewing Policy:**

Affirmed.

**Key Questions For Authors:**

1. Prior work has already shown equivalence between certain TTT formulations and linear attention. What is the precise novelty of the theoretical contribution beyond the empirical results supporting the meorization contradictions?
2. Other concurrent recurrent networks taking on the optimizer perspective may also have similar theoretical connections as the ones presented here. If so, they can also be view as a family of linear attention models. Including such discussion would improve the audience's understanding of the contribution of this work.

**Limitations:**

yes

**Strengths And Weaknesses:**

### Strength
1. The paper provides a useful perspective connecting TTT architectures to linear attention with theoretical supports.
2. Sufficient empirical findings challenge the memorization perspective and the ablations support the linear attention view.
3. The paper is well-written and easy to follow.

### Weakness
1. Prior work (e.g., Sun et al.) has already shown that certain TTT formulations reduce to linear attention. The novelty here appears mainly in extending this observation to broader architectures.
2. As the proposed changes renders itself a class of linear attention variant. It is unclear how it differs to other variants like Mamaba, gated linear attention, deltanet, etc.
3. Clarity issue in Section 5.1. It is unclear whether the ablation steps are applied sequentially or independently.

---

> ### Author Rebuttal · Authors · 2026-03-30
>
> We thank the reviewer for the positive evaluation and for finding the empirical evidence sufficient.
>
> **Q1: Theoretical novelty over original TTT. What is the precise novelty beyond Sun et al. that already showed TTT–LA equivalence?**
>
> Sun et al. showed TTT–LA equivalence only for a single linear layer with zero initialization. This result does not cover any popular TTT variant used in practice: LaCT, ViTTT, and Titans all use multi-layer inner loops with nonlinear architectures, falling outside Sun et al.'s analysis. We believe this gap partly explains why more recent TTT works are typically framed from a memorization perspective rather than through the lens of linear attention.
>
> Our work extends this in two directions:
>
> 1. We identify four empirical anomalies that challenges the widely-believed key–value memorization interpretation of TTT.
> 2. Theorems 1–3 show that even TTT models with arbitrary multi-layer nonlinear inner loops and complex update rules can be interpreted through the lens of linear attention. These analysis apply to most TTT architectures (e.g., LaCT, ViTTT, Titans, and related variants), offering a unified perspective that helps explain the aforementioned empirical observations. They also suggest practical benefits, including a parallel formulation of TTT, which to our knowledge has not been previously demonstrated.
>
> **Q2: Difference with other LA models. How does the simplified TTT differ from Mamba, GLA, DeltaNet, etc.?**
>
> We would like to clarify that our work does not aim to propose a new linear attention variant. Rather, our simplification experiments are to provide empirical support for our linear attention perspective of understanding TTT: the connection between TTT and linear attention is not purely theoretical, but can also be revealed through controlled simplifications in practice.
>
> **Q3: Ablation clarity in Section 5.1. Are the ablation steps applied sequentially or independently?**
>
> Sequential. Each variant builds on the previous: Variant 1 = Baseline + Step 1, ..., Variant 6 = Variant 5 + Step 6, tracing a path from full TTT to standard LA. We will clarify this in the paper.
>
> **Q4: View other recurrent models as LA. Other concurrent recurrent networks taking the optimizer perspective may also have similar theoretical connections. Can they also be viewed as a family of linear attention models?**
>
> We thank the reviewer for this insightful question. We agree that similar connections may exist for other recurrent models. We will expand the discussion in the camera-ready version to more broadly consider these related architectures, and to clarify the extent to which our perspective may extend to them.

---

> > ### Author Rebuttal · Reviewer_k31F · 2026-04-05
> >
> > Thank you for the reply.
> >
> > Follow up question to Q2.
> > I still don't follow how does the simplified TTT compare to the family of parallelizable state-space or rnn models. In particular, the TTT formulation bears similarity to the work of deltanet (for example work by S. Yang et al, Parallelizing Linear Transformers with the Delta Rule over Sequence Length). It would be good to make a comparison between them.

---

> > > ### Author Response · Authors · 2026-04-06
> > >
> > > We appreciate this suggestion and agree that making the connection to works such as DeltaNet more explicit would improve clarity.
> > >
> > > The TTT-equivalent formulation of DeltaNet, as presented in their paper, can be written as:
> > >
> > > $$\\begin{aligned} f(x) &= xW_0, \\\\ \\mathcal{L}(x) &= \\tfrac{1}{2}|f(x) - v|^2, \\end{aligned}$$
> > >
> > > The update process is:
> > >
> > > $$\\begin{aligned} W_1 &= W_0 - \\beta\\nabla_{(W)} \\mathcal{L}(f(k)) \\\\ &= W_0 - \\beta k^\\top (W_0 k - v) \\\\ &= W_0 - \\beta k^\\top v_{0} + \\beta k^\\top v \\end{aligned}$$
> > >
> > > The apply process is:
> > >
> > > $$o = qW_1 = q(W_0 - \\beta k^\\top v_{0} + \\beta k^\\top v)$$
> > >
> > > In contrast, a standard linear attention (our Variant 6) uses Frobenius inner product objective:
> > >
> > > $$\\begin{aligned} f(x) &= xW_0, \\\\ \\mathcal{L} &= \\langle f(k), v \\rangle \\end{aligned}$$
> > >
> > > The update process is:
> > >
> > > $$W_1 = W_0 - \\nabla_{(W)} \\mathcal{L}(f(k)) = W_0 - k^\\top v$$
> > >
> > > The apply process is:
> > >
> > > $$o = qW_1 = q(W_0 - k^\\top v)$$
> > >
> > > The connection between the two has been well studied in both the TTT and linear attention literature. Both correspond to TTT with a single-layer MLP inner loop and a simple gradient descent update rule, which has been shown to admit parallel implementations. Our work builds on this line by extending the connection between TTT and linear attention to more general settings, including multi-layer MLP inner loops and more complex update rules (e.g., momentum, Muon).
> > >
> > > We are happy to incorporate a more explicit discussion of DeltaNet and related models in the revision, if the reviewer feels this would be helpful.

---

### Official Review · Reviewer_by2m · 2026-03-08

**Soundness:** 2
**Presentation:** 3
**Significance:** 2
**Originality:** 2
**Overall Recommendation:** 3
**Confidence:** 3

**Summary:**

The paper argues that a broad class of test-time training (TTT) models should be viewed as learned linear attention rather than test-time memorization. It supports this claim with empirical observations (downstream performance vs. inner-loop fitting, competitive gradient ascent, Q/K mismatch) and a mathematical reformulation. Specifically, it shows that when the inner-loop function has a linear bias-free final layer, the updated model output can be written in a linear-attention form. The appendix provides proofs for various TTT implementations (LaCT, ViTTT) and a parallel formulation.

**Compliance With Llm Reviewing Policy:**

Affirmed.

**Final Justification:**

The paper provides a useful perspective linking TTT to linear attention, and the theoretical reformulations and ablations are informative. The rebuttal improved the empirical case and clarified some technical points, but it did not fully resolve my main concern that the simplification results may reflect over-parameterization or optimization inefficiency rather than a strong equivalence claim. I also still find the framing broader than the technical scope actually established. For these reasons, while I see clear merit in the work, I am keeping my current weak reject score.

**Key Questions For Authors:**

- Can you provide quantitative evidence for the Q/K mismatch (e.g., Cosine Similarity or KL-Divergence between the feature distributions) to supplement the t-SNE plots?
- How much does the "linear attention" interpretation break down if a bias term is introduced or if multiple layers are made dynamic? Is it a "loose" equivalence, or does the mechanism become fundamentally different?

**Limitations:**

This paper discusses the limitations in the conclusion section. Also, this paper does not discuss the societal impact.

**Strengths And Weaknesses:**

### Strengths

- Viewing TTT through the lens of linear attention is a compelling conceptual shift that moves the field away from purely "black-box" optimization interpretations.
- The algebraic reformulations in the appendix (for single-step, unrolled, and momentum cases) appear technically sound and provide a solid formal foundation for the claim.
- The paper includes a comprehensive ablation study (Table 2) that effectively isolates the impact of specific architectural components, such as weight normalization and the number of dynamic layers.


### Weaknesses

- The title and headline claims are significantly broader than the technical results. The main theorem relies on a linear, bias-free final layer, and the paper explicitly restricts its scope to TTT variants with key-value binding losses. While the authors are transparent about these limits in the text, the framing suggests a universal property of TTT that is not established here.
- Much of the "anti-memorization" evidence is suggestive rather than definitive. For example, the Q/K mismatch argument relies heavily on t-SNE visualizations. While visually interesting, t-SNE can be sensitive to hyperparameters and does not provide a quantitative metric of the "mismatch" or "misalignment" between the distributions.
- The paper shows that TTT can be simplified into a parallelizable linear-attention form without losing performance. However, one could interpret this result as suggesting that the original TTT models were simply over-parameterized or poorly optimized, rather than proving that "TTT is linear attention" in its most general form.
- The experimental evidence is concentrated on a specific set of models (LaCT and ViTTT). Expanding the analysis to a wider variety of TTT-based sequence models would strengthen the claim that this is a general property of the paradigm.

---

> ### Author Rebuttal · Authors · 2026-03-30
>
> We thank the reviewer for the careful reading and for recognizing the soundness of the proofs and the ablation study.
>
> **Q1: "TTT is linear attention" overstates. The title/claims are broader than the technical results (requires linear bias-free final layer, key-value binding losses). Clarify scope or adjust framing?**
>
> Please refer to our response to Reviewer GS5b, Q1, where we address this shared concern in detail.
>
> **Q2: Quantitative evidence for distribution mismatch. Provide quantitative evidence for Q/K mismatch (e.g., Cosine Similarity or KL-Divergence) to supplement t-SNE plots.**
>
> Thanks for the suggestion. To provide additional evidence that the TTT mechanism does not perform retrieval using the Q/K space as suggested by prior work, we supplement the t-SNE visualizations with five quantitative distribution-comparison metrics. The table below reports these metrics between Q & K and V & O at layers 1, 4, and 7 (matching the t-SNE panels in the paper):
>
> We report five metrics — Cosine Similarity (Cos Sim), Centered Kernel Alignment (CKA), KL-Divergence (KL-Div), Maximum Mean Discrepancy (MMD), and Wasserstein Distance — where ↑ means higher is more aligned and ↓ means lower is more aligned:
>
> | Pair | Layer | Cos Sim ↑ | KL-Div ↓ | MMD ↓ | CKA ↑ | Wasserstein ↓ |
> |------|-------|-----------|----------|-------|-------|---------------|
> | Q vs K | 1 | 0.163 | 3.55 | 0.708 | 0.003 | 0.021 |
> | Q vs K | 4 | 0.078 | 2.24 | 0.577 | 0.005 | 0.022 |
> | Q vs K | 7 | 0.011 | 1.81 | 0.504 | 0.013 | 0.022 |
> | V vs O | 1 | −0.010 | 2.5e5 | 0.564 | 0.005 | 0.373 |
> | V vs O | 4 | 0.008 | 9.1e4 | 0.502 | 0.012 | 0.306 |
> | V vs O | 7 | 0.019 | 5.7e4 | 0.469 | 0.012 | 0.314 |
>
> Across all layers, cosine similarity between Q and K stays below 0.17 and CKA stays below 0.014—both near zero, confirming that the two representations occupy nearly orthogonal subspaces. KL-Divergence, MMD, and Wasserstein distance all remain large, consistent with a substantial distributional gap. V vs. O shows an even larger gap, with KL-Divergence several orders of magnitude higher due to the scale disparity between V and O.
>
> **Q3: Original TTT over-parameterized or poorly optimized. Could the simplification results simply indicate over-parameterization rather than proving equivalence?**
>
> Please refer to our response to Reviewer qYky, Q2(a), where we address this point in detail.
>
> **Q4: Wider variety other than LaCT and ViTTT. Can the analysis be extended to other TTT-based sequence models?**
>
> Our theorems apply broadly to TTT architectures based on key–value binding. We chose LaCT and ViTTT as primary case studies because they are representative methods with open-source implementations. Other TTT variants, such as Titans and Atlas, also satisfy the assumptions required by our analysis, and we expect our results to extend to them as well. However, as these models are not publicly available, we were unable to include empirical evaluations. We will note this as an avenue for future validation.
>
> **Q5: What if bias term is introduced or MLP layers are dynamic? How much does the LA interpretation break down?**
>
> For the bias case, consider $f_t(x) = \\phi_t(x)\\,W_t + b_t$. The gradient of the loss $\\mathcal{L}$ w.r.t. the bias is $\\nabla_{b_t}\\mathcal{L} = \\frac{\\partial\\mathcal{L}}{\\partial f_t(k)}$, so after one gradient step both $W$ and $b$ are updated:
>
> $$\\begin{aligned} W_{t+1} &= W_t + \\phi_t(k)^\\top g_t(k), \\\\ b_{t+1} &= b_t + g_t(k), \\end{aligned}$$
>
> where $g_t(k)=-\\eta\\frac{\\partial\\mathcal{L}}{\\partial f_t(k)}$ as in Theorem 1. Unrolling over tokens $0,\\dots,t$ and evaluating on query $q_t$:
>
> $$o_t = \\phi_{t+1}(q_t)\\!\\left(W_0 + \\sum_{i=0}^{t} \\phi_i(k_i)^\\top g_i(k_i)\\right) + b_0 + \\sum_{i=0}^{t} g_i(k_i).$$
>
> The first term is the linear attention form from Theorem 2. The second term $\\tilde{b}_t$ depends only on the key–value history, not on the query $q_t$. This bias term can be absorbed into linear attention by augmenting the feature map: define $\\bar\\phi(x)=[\\phi(x),\\;1]\\in\\mathbb{R}^{D_h+1}$ and the initial state $\\bar S_0=[W_0;\\;b_0]\\in\\mathbb{R}^{(D_h+1)\\times D_v}$. Then:
>
> $$o_t = \\bar\\phi_{t+1}(q_t)\\!\\left(\\bar S_0 + \\sum_{i=0}^{t} \\bar\\phi_i(k_i)^\\top g_i(k_i)\\right),$$
>
> which is again a linear attention form with feature dimension $D_h+1$. The linear attention interpretation therefore holds still; the only change is one extra feature dimension.
>
> For multiple dynamic layers, our interpretation also holds: Theorems 5.1–5.3 do not assume updates are restricted to the last layer. Updates to earlier layers are absorbed into the dynamic $\\phi$ function, which acts as a learned nonlinear kernel in the resulting operator.
>
> **Q6: Missing societal impact discussion.**
>
> We will revise the camera ready version with the following impact statement: Our work simplifies and parallelizes existing architectures, potentially reducing computational cost and energy consumption.

---

> > ### Author Rebuttal · Reviewer_by2m · 2026-04-04
> >
> > The rebuttal is helpful but only partially resolves my concerns. It improves the empirical support, but it still does not convincingly address whether the observed simplifications reflect genuine equivalence or simply over-parameterization / optimization inefficiency in existing TTT designs. More importantly, the central claim remains overstated relative to the paper’s actual technical scope, so substantial revision is still needed to align the framing with what is truly shown. For these reasons, I will keep my current score.

---

> > > ### Author Response · Authors · 2026-04-04
> > >
> > > We thank you for the follow-up and for clarifying what remains open after the rebuttal.
> > >
> > >
> > > **Re: “whether the observed simplifications reflect genuine equivalence or simply over-parameterization / optimization inefficiency in existing TTT designs”**
> > >
> > >
> > > The simplification experiments could indeed be interpreted as suggesting over-complexity in current TTT designs. However, this is not the primary claim or focus of our work. As noted in the rebuttal, these experiments are intended as empirical evidence supporting the connection between TTT and linear attention.
> > > Specifically, for a complex TTT design, we demonstrate a trajectory along which it can be systematically reduced to a standard linear attention (LA) formulation. Importantly, each intermediate variant along this trajectory admits a corresponding linear-attention interpretation (e.g., the transition from a multilayer to a single-layer inner model can be viewed as removing the kernel function from the LA perspective).
> > > Taken together, we view this simplification trajectory as evidence of a genuine equivalence between TTT and LA, rather than merely an indication of over-parameterization. If the reviewer finds it helpful, we would be happy to include a more detailed discussion of these ablations in the revision.
> > >
> > >
> > > **Re: “the central claim remains overstated relative to the paper’s actual technical scope”**
> > >
> > >
> > > We agree that clarity of scope is important. In the rebuttal, we proposed revising the title from “TTT” to “key–value binding TTT” to better reflect the precise setting we analyze.
> > > Our claims have consistently focused on key–value binding TTT variants, which, to the best of our knowledge, all employ a linear, bias-free final layer (we would greatly appreciate any pointers to counterexamples). We have also included an additional argument in the rebuttal showing that our analysis naturally extends to the case where the final layer includes a bias term.
> > > That said, we are happy to further refine the wording throughout the paper to better align with the intended scope. If the reviewer has specific suggestions beyond the title, we would greatly appreciate the guidance.

---

### Official Review · Reviewer_GS5b · 2026-03-08

**Soundness:** 3
**Presentation:** 3
**Significance:** 2
**Originality:** 2
**Overall Recommendation:** 4
**Confidence:** 3

**Summary:**

The paper reframes TTT as learned linear attention rather than online meta-learning/memorisation. The authors present four empirical anomalies contradicting the memorisation view, formal proofs that TTT with gradient descent reduces to linear attention, an ablation trajectory simplifying TTT architectures, and a parallel formulation achieving greater TTT-layer throughput.

**Compliance With Llm Reviewing Policy:**

Affirmed.

**Final Justification:**

The paper makes a useful contribution by extending the linear attention equivalence to more general TTT architectures. The rebuttal fully addressed my concerns. My weak accept reflects the fact that the core insight builds directly on prior work showing the equivalence in a more restricted case, which limits the originality and significance of the contribution.

**Key Questions For Authors:**

N/A

**Limitations:**

Please include a brief societal impact statement in accordance with ICML guidelines, addressing potential broader consequences of more efficient sequence modelling.

**Strengths And Weaknesses:**

**Strengths**

The paper makes a useful contribution by extending the linear attention equivalence to nonlinear TTT, providing a clean ablation trajectory, and demonstrating practical parallelisation benefits when certain simplifications are made. The four identified anomalies in the TTT methods provide strong, concrete motivation for rethinking the memorisation narrative. The paper also provides clean and easy-to-follow proofs.

**Weaknesses**
- The title and framing suggest that existing TTT architectures are essentially linear attention mechanisms. However, the empirical contribution of the paper appears closer to showing that current TTT architectures can be progressively simplified into standard linear attention with minimal performance loss. This is a valuable result, but it differs from the stronger claim implied by the title. Clarifying this distinction would improve the presentation and help readers better understand the practical implications of the work.
- Since Variant 6 is equivalent in structure to standard linear attention, the most natural baseline is a linear attention model trained directly from scratch under identical conditions. Without this, the minimal performance loss claimed by the authors is only measured relative to the original TTT formulation. If Variant 6 performs the same as natively trained linear attention, the TTT framing adds nothing practical; if it performs better, that would be a surprising and important finding that the paper is currently missing the opportunity to demonstrate.
- The introduction attributes enhanced expressiveness of their proposed simplified TTT designed to the learned nonlinear kernel, but this is neither formally proven nor empirically demonstrated. The paper's own ablation contradicts it: moving from the nonlinear MLP kernel to a single linear layer (Variant 2→3) costs negligible performance, and Variant 6, which removes the learned kernel entirely, performs nearly as well as the full model. The claim should either be removed or supported with dedicated evidence.

---

> ### Author Rebuttal · Authors · 2026-03-30
>
> We thank the reviewer for the positive assessment, and in particular for noting the strong motivation from the four anomalies.
>
> **Q1: "TTT is linear attention" overstates. The result is closer to "TTT can be progressively simplified into linear attention." Can the framing be clarified?**
>
> We appreciate and agree with the reviewer's suggestion. We are happy to revise the presentation to be more precise. In particular, we will (1) scope our discussion to key–value binding forms of TTT (excluding works such as TTT-E2E), and (2) emphasize that TTT can be interpreted from the perspective of linear attention, rather than asserting that it "is linear attention".
>
> **Q2: Missing standard linear attention baseline. Variant 6 should be compared against a linear attention model trained from scratch under identical conditions.**
>
> Variant 6 in our ablation is TTT with single layer inner loop, which *is* exactly standard linear attention. To make this explicit, the TTT formulation of Variant 6 can be written as follows.
>
> The TTT inner loop function and loss of Variant 6 is:
>
> $$\\begin{aligned} f(x) &= xW_0 \\\\ \\mathcal{L} &= \\langle f(k), v \\rangle \\end{aligned}$$
>
> The update process is:
>
> $$W_1 = W_0 - \\nabla_{(W)} \\mathcal{L}(f(k)) = W_0 - k^\\top v$$
>
> The apply process is:
>
> $$o = qW_1 = q(W_0 - k^\\top v)$$
>
> This recovers the standard linear attention form (up to a sign difference that can be absorbed into the parameterization of $v$). Therefore, Variant 6 already serves as a linear attention baseline under the same training setup, without requiring a separately trained model.
>
> **Q3: Claim contradiction. The learned nonlinear kernel expressiveness claim is contradicted by the ablation (Variant 2→3 and Variant 6). Substantiate or remove?**
>
> We thank the reviewer for pointing this out. We do not intend to claim novelty in kernel design. Rather, we report an empirical observation: the full model performs slightly better than Variant 6 on both LLM (+0.87 perplexity) and NVS (+0.24 dB). We will revise the introduction to present this more cautiously as an empirical observation, rather than a strong claim about expressiveness, and ensure the wording is consistent with the ablation results.
>
> **Q4: Missing societal impact statement.**
>
> We will revise the camera ready version with the following impact statement: Our work simplifies and parallelizes existing architectures, potentially reducing computational cost and energy consumption.

---

> > ### Author Rebuttal · Reviewer_GS5b · 2026-04-01
> >
> > Thank you for your response. My concerns have been fully resolved, and I will maintain my current positive score.

---

> > > ### Author Response · Authors · 2026-04-04
> > >
> > > Thank you again for your thoughtful response and for taking the time to review our paper. We're glad to hear that your concerns have been addressed. If any further questions arise, please don’t hesitate to reach out—we’d be happy to continue the discussion.
> > > If our response has sufficiently addressed your concerns, we would be grateful if you would consider revisiting the score. We truly appreciate your time and effort in helping to improve our work.

---

### Official Review · Reviewer_qYky · 2026-03-15

**Soundness:** 3
**Presentation:** 3
**Significance:** 2
**Originality:** 2
**Overall Recommendation:** 4
**Confidence:** 2

**Summary:**

This paper claims that some common sense about understanding  test-time-training as a form of meta learning  that memorizes the key-value mapping at test time is not accurate, through some experiments like QK distribution mismatch, gradient ascent. And systematically shows that TTT algorithm can be simplified as linear attention

**Compliance With Llm Reviewing Policy:**

Affirmed.

**Key Questions For Authors:**

see S and W

**Limitations:**

The authors does not include the impact statement

**Strengths And Weaknesses:**

W:
1. In general I think the claim of TTT is memorizing key-value mapping rather than linear attention is not some common sense. I believe many researchers still know the strong connection between linear attention and TTT, as the original TTT-MLP paper claimed. This somewhat reduce the originality of the paper.
2. Though paper shows that cleaning some techniques in LaCT can increase recurrency, it still not cover the entire performance, seems gradient orthogonalization still performs well. And seems not include scaling experiments, don't know if change the model size (in the same arch) would make the gap larger, which I think can make the experiments more solid.
3. Seems no impact statement

S:
1. Writing is good, easy to follow
2. The qualitative experiments are interesting, like the gradient ascent one, showing that the inner loop is kind of tricky. And from theoretical aspect, update the last linear layer but keep the kernel frozen may be more similar to linear attention, while the experiments support this, which is nice
3. The ablation studies of the method clearly show the effects of the components, like only update the last linear layer can be beneficial. I think though this paper does not propose some new techniques directly, this kind of cleaning ablation studies can be helpful for community to understand the architectures.

---

> ### Author Rebuttal · Authors · 2026-03-30
>
> We thank the reviewer for the positive evaluation and for noting that the ablation studies can help the community understand these architectures.
>
> **Q1: TTT is memorizing KV mapping is not common sense and "TTT = linear attention" is not a novel claim. The connection is already known from the original TTT-MLP paper. How does this work go beyond that?**
>
> To our best knowledge, the memorization-based interpretation of TTT remains the dominant framing in the TTT community. For example,
>
> 1. The original TTT (Sun et al., §2.1) defines the update rule as optimizing $\\lVert f(\\theta_K x;W)-\\theta_V x\\rVert^2$, a key-to-value reconstruction loss.
>
> 2. LaCT (Zhang et al., §2.1) states the objective is "to encode the KV cache into a neural memory with fixed state size as *accurate* as possible."
>
> 3. Titans (Behrouz et al., title & §3.1) is titled "Learning to *Memorize* at Test Time" and writes "the model learns how to memorize the mapping between keys and values at test time."
>
> 4. ViT³ (Han et al., §3.2) says TTT "compresses $K,V$ into the module weights $W$ through a few self-supervised online training steps."
>
> 5. Miras (Behrouz et al., Def. 3.1) formalizes all such models as associative memories $\\mathcal{M}\\!:\\!\\mathcal{K}\\!\\to\\!\\mathcal{V}$.
>
> 6. VGG-T³ (Elflein et al., §3.1) states "this optimization embeds the mapping from keys $k_i$ to values $v_i$ into a learnable network."
>
> 7. Atlas (Behrouz et al., §1) explicitly adopts the term "test time memorization" and states the process involves "storing and retrieving information strictly within the global context."
>
> 8. Nested Learning (Behrouz et al., abstract) describes optimizers as "associative memory modules that aim to compress the gradients' information."
>
> We welcome any pointers to perspectives we may have missed.
>
> Sun et al. showed TTT–LA equivalence only for a single linear layer with zero initialization. This result does not cover any popular TTT variant used in practice: LaCT, ViTTT, and Titans all use multi-layer inner loops with nonlinear architectures, falling outside Sun et al.'s analysis. We believe this gap partly explains why more recent TTT works are typically framed from a memorization perspective rather than through the lens of linear attention.
>
> Our work extends this in two directions:
>
> 1. We identify four empirical anomalies that challenges the widely-believed key–value memorization interpretation of TTT.
> 2. Theorems 1–3 show that even TTT models with arbitrary multi-layer nonlinear inner loops and complex update rules can be interpreted through the lens of linear attention. These analysis apply to most TTT architectures (e.g., LaCT, ViTTT, Titans, and related variants), offering a unified perspective that helps explain the aforementioned empirical observations. They also suggest practical benefits, including a parallel formulation of TTT, which to our knowledge has not been previously demonstrated.
>
> **Q2: Simplified model doesn't match performance of original model. (a) Gradient orthogonalization still performs well and the gap is not fully closed. (b) Do scaling experiments (varying model size) make the gap larger?**
>
> (a) The goal of these ablations is not to match the full model's performance. Rather, they are designed to illustrate a systematic path that reduces a complex TTT model to vanilla linear attention (Variant 6). Along this trajectory, each intermediate variant admits a corresponding linear-attention interpretation—for example, Variant 3 (which reduces a multi-layer MLP to a single linear layer) can be viewed as removing the kernel function in linear attention. Taken together, these results provide empirical support for our new perspective: the connection between TTT and linear attention is not purely theoretical, but can also be revealed through controlled simplifications in practice.
>
> (b) Our current evaluation already involves substantial compute (LLM: 3,136 A100-hrs; NVS: 1,064 A100-hrs; ViTTT: 224 H100-hrs). Extending these experiments to multiple additional scales across all variants would significantly increase the total cost. More importantly, our claim is not that TTT's additional components are redundant at all scales, but rather that TTT's underlying mechanism can be understood through the lens of linear attention, and that there exists a clear path to systematically ablate this complex formulation into its simplest form. Studying scaling behavior is certainly an interesting direction, but we view it as largely orthogonal to our current analysis. That said, we believe our perspective of TTT as linear attention could provide a useful framework for guiding such future investigations.
>
> **Q3: Missing impact statement.**
>
> We will revise the camera ready version with the following impact statement: Our work simplifies and parallelizes existing architectures, potentially reducing computational cost and energy consumption.

---

> > ### Author Rebuttal · Reviewer_qYky · 2026-04-04
> >
> > Thanks authors for the rebuttal, most of my concerns are resolved. I would keep my positive score

---

> > > ### Author Response · Authors · 2026-04-04
> > >
> > > Thank you again for your thoughtful response and for taking the time to review our paper. We're glad to hear that your concerns have been addressed. If any further questions arise, please don’t hesitate to reach out—we’d be happy to continue the discussion.
> > > If our response has sufficiently addressed your concerns, we would be grateful if you would consider revisiting the score. We truly appreciate your time and effort in helping to improve our work.

---

### Decision · Program_Chairs · 2026-04-30

**Decision:**

Accept (regular)

**Comment:**

The paper demonstrates that a broad class of TTT architectures, including multi-layer nonlinear variants like LaCT, ViTTT, and Titans, can be expressed as learned linear attention operators. This meaningfully extends the prior equivalence result by Sun et al., which covered only the single linear layer case that no practical TTT variant actually uses. The four empirical anomalies challenging the dominant memorization interpretation are well-constructed and provide strong motivation, and the theoretical contribution (Theorems 1–3) is clean with no correctness issues identified. Three of four reviewers are satisfied post-rebuttal (4/4/4), and the holdout's concern (score 3) centers on whether the simplifications reflect genuine equivalence or over-parameterization — a reasonable interpretive question but not a technical flaw. The title oversells the scope, and the authors have agreed to narrow it to key-value binding TTT variants; the camera-ready should follow through on this and clearly delineate the boundary with Sun et al.'s prior result. The practical benefit of a parallel TTT formulation and the systematic ablation trajectory from full TTT to vanilla linear attention add concrete value beyond the conceptual reframing.